# High Fructose and High Fat Diet Impair Different Types of Memory through Oxidative Stress in a Sex- and Hormone-Dependent Manner

**DOI:** 10.3390/metabo12040341

**Published:** 2022-04-12

**Authors:** Edwin Chávez-Gutiérrez, Claudia Erika Fuentes-Venado, Lorena Rodríguez-Páez, Christian Guerra-Araiza, Carlos Larqué, Erick Martínez-Herrera, María Esther Ocharan-Hernández, Joel Lomelí, Marco A. Loza-Mejía, Juan Rodrigo Salazar, Dulce María Meneses-Ruiz, Juan Manuel Gallardo, Rodolfo Pinto-Almazán

**Affiliations:** 1Doctorado en Ciencias en Biomedicina y Biotecnología Molecular, Escuela Nacional de Ciencias Biológicas, Instituto Politécnico Nacional, Mexico City 11340, Mexico; chz_edwin.bioexp@hotmail.com; 2Servicio de Medicina Física y Rehabilitación, Hospital General de Zona No 197, Texcoco 56108, Mexico; cefvenado@hotmail.com; 3Departamento de Bioquímica, Escuela Nacional de Ciencias Biológicas, Instituto Politécnico Nacional, Prolongación Manuel Carpio y Plan de Ayala s/n, Col. Santo Tomás, Miguel Hidalgo, CP 11340, CDMX, Mexico City 11340, Mexico; lorena_rpaez@yahoo.com.mx; 4Medical Research Unit in Pharmacology, Specialities Hospital Bernardo Sepúlveda, National Medical Center XXI Century, Social Security Mexican Institute (IMSS), Av. Cuauhtémoc 330, Mexico City 06720, Mexico; 5Departamento de Embriología y Genética, Facultad de Medicina, Universidad Nacional Autónoma de México, Avenida Universidad # 3000, Col. Ciudad Universitaria, Alcaldía de Coyoacán, Mexico City 04510, Mexico; skiuty@hotmail.com; 6Sección de Estudios de Posgrado e Investigación, Escuela Superior de Medicina, Instituto Politécnico Nacional, Mexico City 11340, Mexico; erickmartinez_69@hotmail.com (E.M.-H.); estherocharan@hotmail.com (M.E.O.-H.); jlomeli_glez@yahoo.com.mx (J.L.); 7Efficiency, Quality, and Costs in Health Services Research Group (EFISALUD), Galicia Sur Health Research Institute (IIS Galicia Sur), SERGAS-UVIGO, 36213 Vigo, Spain; 8Design, Isolation, and Synthesis of Bioactive Molecules Research Group, Chemical Sciences School, Universidad La Salle-México, Benjamín Franklin 45, Mexico City 06140, Mexico; marcoantonio.loza@lasalle.mx (M.A.L.-M.); juan.salazar@lasalle.mx (J.R.S.); 9Noncommunicable Diseases Research Group, Universidad La Salle-México, Benjamín Franklin 45, Mexico City 06140, Mexico; dulce.meneses@lasalle.mx; 10Unidad de Investigación Médica en Enfermedades Nefrológicas, Hospital de Especialidades, Centro Médico Nacional Siglo XXI, Instituto Mexicano del Seguro Social, Mexico City 06720, Mexico; jmgallardom@gmail.com; 11Doctorado en Ciencias Biológicas y de la Salud, Universidad Autónoma Metropolitana, Mexico City 14387, Mexico

**Keywords:** metabolic syndrome, ovariectomy, orchiectomy, memory, oxidative stress

## Abstract

Metabolic syndrome (MetS) contributes to the spread of cardiovascular diseases, diabetes mellitus type 2, and neurodegenerative diseases. Evaluation of sex- and hormone-dependent changes in body weight, blood pressure, blood lipids, oxidative stress markers, and alterations in different types of memory in Sprague–Dawley rats fed with a high fat and high fructose (HFHF) diet were evaluated. After 12 weeks of feeding the male and female rats with HFHF, body weight gain, increase in blood pressure, and generation of dyslipidemia compared to the animals fed with chow diet were observed. Regarding memory, it was noted that gonadectomy reverted the effects of HFHF in the 24 h novel object recognition task and in spatial learning/memory analyzed through Morris water maze, males being more affected than females. Nevertheless, gonadectomy did not revert long-term memory impairment in the passive avoidance task induced by HFHF nor in male or female rats. On the other hand, sex-hormone–diet interaction was observed in the plasma concentration of malondialdehyde and nitric oxide. These results suggest that the changes observed in the memory and learning of MetS animals are sex- and hormone-dependent and correlate to an increase in oxidative stress.

## 1. Introduction

Metabolic syndrome (MetS) constitutes a global public health problem, considering that it is estimated that a quarter of the world’s population suffers from it [1]. The metabolic deregulations associated with this condition include central obesity, insulin resistance, arterial hypertension, dyslipidemia (hypercholesterolemia and hypertriglyceridemia), and glucose intolerance [1,2]. The visceral adipose tissue has been identified as a key factor in the pathogenesis of MetS since it causes an increase in systemic oxidative stress (OS), endothelial dysfunction, and excessive production of adipokines, leading to other comorbidities [2,3].

Different emotional and behavioral alterations have been reported as long-term secondary problems associated with MetS. Likewise, the high consumption of high-fat and fructose diets has also been correlated with both emotional and behavioral alterations, as well as MetS [4,5,6,7].

On the other hand, previous reports indicate that sex differences have an impact in MetS, which is manifested in the glycemic index, body fat distribution, size and function of the adipocytes, hormonal regulation of body weight, and adiposity, as well as in estrogen levels in the clustering of risk factors [8]. Regarding behavioral aspects, an increase in anxiety, cognitive impairment, and alterations on the brain’s structure in animal models fed with hypercaloric diets have been reported [9,10]. Interestingly, Gancheva et al. reported an association between the increase in serum lipid peroxidation (LPO) and the increase in anxiety and depression-like behavior after three months of feeding with a high-fat diet, which confirms the relevance of the OS in MetS, as well as on the behavioral effects [11]. 

However, even when there are multiple studies in animal models in which the effects of hypercaloric diets on memory and learning have been analyzed, few of them have assessed the sexual dimorphism of these effects. In these studies, the tests were carried out on young intact animals in which it was found that female rats showed a lower susceptibility to developing MetS and neurocognitive impairment when exposed to high-fat diets [12,13]. In accordance with these findings, in a previous study, sex differences were reported in the performance of spatial learning/memory test, in short- and long-term memory, as well as the decrease in sex hormones in animals fed during 12 weeks with a diet rich in fat and fructose (HFHF) [14]. Considering that such studies were performed in young animals, the lower susceptibility to cardiovascular and neuronal damages produced by hypercaloric diets observed in female animals may be attributed to the effects of sex hormones [12,13,14,15]. 

The sex hormones, such as progesterone (P4), estradiol (E2), and testosterone (T), are involved in the regulation of many processes, ranging from the maintenance of tissue growth to reproduction. These processes can be mediated through the activation of nuclear receptors, triggering genomic events, as well as through nongenomic or rapid-action mechanisms, such as those that depend on nitric oxide (NO), including vasodilation, ischemic myocardial damage, response to endothelial damage, and relaxation of the coronary artery, among others [16]. In the specific case of E2, another nongenomic effect is caused directly by its chemical structure, as it possesses a phenolic ring which can donate hydrogen atoms to peroxilipidic radicals, reactive oxygen species (ROS), and other harmful radicals, which may inhibit the LPO [17]. 

For all the above, the aim of the present study was to analyze the sexual dimorphism and the effects of hormones in cognitive task performance and hippocampal functions of Sprague–Dawley rats fed with an HFHF diet. For this purpose, three different tests were used: novel object recognition (NOR), Morris water maze (MWM), and the passive avoidance test (PAT) and their possible correlation with the changes in oxidative stress markers (malondialdehyde (MDA) and nitric oxide (NO)) was evaluated. 

## 2. Results

### 2.1. Determination of Diet and Sex-Dimorphism Effects on Somatometric and Metabolic Parameters

In regards of food intake, the main effects of both diet exposure (F (1, 72) = 31.53, *p* < 0.0001) and sex (F (1, 72) = 18.83, *p* < 0.0001) were observed. Moreover, significant interactions between sex and HFHF diet exposure (diet exposure × sexual interaction: F (1, 72) = 33.78, *p* < 0.0001) and between sex and hormones (Sex × hormones interaction: F (1, 72) = 26.53, *p* < 0.0001) were detected. Feeding with the hypercaloric diet did not result in an increase in food consumption by either HFHF or HFHF-fed orchiectomized males (HFHF-ORCH) rats, unlike those observed in female rats, in which hypercaloric diet induced a higher food consumption in the HFHF-F and HFHF-fed ovariectomized females (HFHF-OVX) rats compared to the control females (CF) group. However, all groups fed with the hypercaloric diet consumed more kcal than those fed with the standard diet (main effect of diet exposure in Kcal/g: F (1, 72) = 163.3, *p* < 0.0001). Moreover, orchiectomy significantly decreased the food intake in the ORCH group, producing a non-statistically significant increase in kcal/gr consumption compared to the control males (CM) group. In female groups, OVX produced an increase in kcal consumption in the HFHF-OVX group compared to CF and HFHF-F groups (Appendix A). 

In terms of body weight, some main effects from diet exposure (F (1, 72) = 36.03 *p* < 0.0001); sex (F (1, 72) = 43.48 *p* < 0.0001); hormones (F (1, 72) = 3.095, *p* = 0.0830); and interaction of sex and hormones (F (1, 72) = 55.54, *p* < 0.0001) were observed. HFHF-M animals showed the highest body weight gain compared to the other male groups (main effect of diet exposure: F (1, 72) = 78.27, *p* < 0.0001; main effect of hormones: F (1, 72) = 68.73, *p* < 0.0001; interaction of diet exposure and hormones: F (1, 72) = 1.748, *p* < 0.0074). In addition, HFHF-ORCH rats (main effect of diet exposure: F (1, 72) = 78.27, *p* < 0.0001) also showed a higher body weight gain compared to ORCH animals. However, hormones induced a lower weight gain in ORCH (main effect of hormones: F (1, 72) = 68.73, *p* < 0.0001) and HFHF-ORCH (main effect of hormones: F (1, 72) = 68.73, *p* < 0.0001; interaction of diet exposure and hormones: F (1, 72) = 1.748, *p* < 0.0074) groups compared to CM rats (Figure 1A). After 12 weeks of diet administration, the HFHF-OVX group showed the highest body weight gain compared to the other female groups (main effect of diet exposure: F (1, 72) = 135.4, *p* < 0.0001; main effect of hormones: F (1, 72) 121.4, *p* < 0.0001; interaction of diet exposure and hormones: F (1, 72) = 5.199, *p* < 0.0001). Additionally, we noted a higher weight gain in HFHF-F (main effect of diet exposure: F (1, 72) = 135.4, *p* < 0.0001) and OVX (main effect of hormones: F (1, 72) 121.4, *p* < 0.0001) rats compared to the CF animals. However, no difference was observed between HFHF-F and OVX groups (Figure 1B).

Upon assessing sex differences in weight change, CM and HFHF-M groups had a higher body weight gain than CF and HFHF-F groups since the third week (main sex effect: F (1, 72) = 72.15, *p* < 0.0001; interaction of diet exposure × sex effects: F (1, 72) = 5.427, *p* < 0.0001) (Figure 1C). Furthermore, no difference in weight gain was observed between HFHF-ORCH and HFHF-OVX groups, although both groups showed increased weight gain when compared to ORCH and OVX groups (main effect of diet exposure: F (1, 72) = 38.78, *p* < 0.0001; main sex effect: F (1, 72) = 87.99, *p* < 0.0001) (Figure 1D).

Assessment of blood pressure showed higher levels of systolic blood pressure in rats of HFHF-M, HFHF-ORCH, HFHF-F, and HFHF-OVX groups compared with the CM, ORCH, CF, and OVX groups (main effect of dietary exposure: F (1, 72) = 134.6, *p* < 0.0001). In addition, a higher increase was observed in HFHF-M rats compared to HFHF-F (main sex effect: F (1, 72) = 9.277, *p* < 0.0034). Moreover, there was a statistically significant increase in HFHF-OVX and HFHF-ORCH compared to CF and CM (interaction of diet exposure, sex, × hormones effects: F (1, 72) = 17.96, *p* < 0.0001) (Figure 1E). On the other hand, hypercaloric diet induced hypertriglyceridemia in HFHF-M, HFHF-ORCH, HFHF-F, and HFHF-OVX groups when compared with control-fed groups (CM, ORCH, CF, and OVX) (main effect of diet exposure: F (1, 72) = 97.22, *p* < 0.0001; main effect of hormones: F (1, 72) = 10.36, *p* < 0.05). No statistical significance was proved due to sex differences (F (1, 72) = 0.1162). However, gonadectomized- and HFHF-fed animals (HFHF-ORCH and HFHF-OVX) showed lower triacylglycerides levels compared to HFHF-M and HFHF-F (sex × hormones interaction was observed: F (1, 72) = 4.499, *p* < 0.05; and diet exposure × hormone interaction: F (1, 72) = 18.57, *p* < 0.001) (Figure 1F). Finally, cholesterol levels increased in the HFHF-M and HFHF-ORCH groups when compared with CM and ORCH groups, respectively; and HFHF-F and HFHF-OVX groups compared with the CF group (main effect of dietary exposure: F (1, 72) = 29.59, *p* < 0.0001; main effect of sex: F (1, 72) = 30.48, *p* < 0.0001). It should be emphasized that that OVX and HFH-OVX animals had higher levels of TC compared to CF rats (main effect of dietary exposure: F (1, 72) = 29.59 and main effect hormones: F (1, 72) = 18.15, *p* < 0.0001) (Figure 1G).

### 2.2. Determination of the Effects of the Hypercaloric Diet on Learning and Memory

#### 2.2.1. Determination of the Effects of the Hypercaloric Diet on the Object Recognition Memory by Novel Object Recognition Task

As expected, at the short-term recognition memory assessment, animals from all groups spent more time exploring the new versus the familiar object. In females, a decrease in the recognition index between the HFHF-OVX group was observed compared to CF, HFHF-F, and OVX groups (*p* < 0.05), as well as an increase in the recognition index of ORCH compared to CM and HFHF-M groups (*p* < 0.001) (main effect of time exposure to the diet (F (1, 72) = 10.88, *p* < 0.0017), sex × hormones interaction (F (1, 72) = 25.65, *p* < 0.0001), and diet × hormones interaction (F (1, 72) = 4.280, *p* < 0.05). Interestingly, a statistically significant difference was observed between the HFHF-M vs. HFHF-F group, with a decrease in the recognition memory index only occurring in the HFHF-M group (F (1, 36) = 30.48, *p* < 0.0001) (Figure 2A).

The analysis of the long-term recognition memory revealed that the HFHF-M group showed a statistically significant decrease in recognition index compared to CM and HFHF-ORCH groups (main effect of diet exposure: F (1, 72) 6.224, *p* = 0.05; diet × hormones interaction: F (1, 36) = 4.454, *p* < 0.05). In addition, MetS induced a reduction in the recognition index in HFHF-OVX (*p* < 0.05) compared to all other female groups (diet × hormones interaction: F (1,36) = 6.554, *p* < 0.05) and the HFHF-ORCH group (sex × diet × hormones interaction: F (1, 72) = 22.00, *p* < 0.0001) (Figure 2B).

#### 2.2.2. Determination of the Effects of the Hypercaloric Diet on Spatial Learning and Memory by Morris Water Maze

Regarding the analysis of spatial memory, it was observed that the HFHF-M group had a statistically significant increase in escape latency when compared to CM, ORCH, and HFHF-ORCH groups (*p* < 0.05) and without statistically significant difference between the other male groups (Figure 3A). Moreover, a statistically significant reduction in escape latency was observed on the 6th day in HFHF-OVX compared to CF, OVX, and HFHF-F groups, without differences between CF and HFHF-F on the 6th day (Figure 3B) (Diet × hormones interaction: 1st day F (1, 72) = 5.346, *p* < 0.05, 6th day F (1, 72) = 3.289, *p* < 0.05, diet: 4th day F (1, 72) = 4.060, *p* < 0.05, 6th day F (1, 72) = 10.02, *p* < 0.05), hormones: 4th day F (1, 72) = 3.991, *p* < 0.05, 6th day F (1, 72) = 30.02, *p* < 0.0001). Sex differences were observed between CM and CF from the 2nd until the 6th day (*p* < 0.05) and between ORCH and OVX (*p* < 0.05), without differences in the escape latency between HFHF-M and HFHF-F groups and between HFHF-ORCH and HFHF-OVX (main effect of sex: 4th day F (1, 72) = 5.173, *p* < 0.05, 6th day F (1, 72) = 4.060, *p* < 0.05, sex × dietary interaction: 4th day F (1, 72) = 9.482, *p* < 0.05, 6th day F (1, 72) = 11.93, *p* < 0.001) (Figure 3C,D).

Furthermore, sex differences were recognized between females and males in the transfer trial (diet × sex × hormone interaction: F (1, 51) = 14.36, *p* = 0.001). CM males passed through the target place more times compared to CF females (main effect of sex: F (1, 72) = 5.796, *p* < 0.05). The HFHF-M group showed a lower number of crossings through the former platform location during the transfer trial compared to CM (*p* < 0.05). In contrast, the HFHF-F group had more crossings through the former platform compared to CF (*p* < 0.05) (diet × sex interaction: F (1, 72) = 11.48, *p* < 0.01). In addition, OVX rats had a higher number of passes compared to CF rats (main effect of hormones: F (1, 72) = 6.534, *p* < 0.05) (Figure 3E). In Figure 3F, the swimming patterns of the eight experimental groups in the transfer trial are observed, starting from the northwest quadrant. The CM, ORCH, and HFHF-ORCH groups traveled a shorter distance and spent more time in the quadrant where the platform was located. In comparison with the CM group, the HFHF-M group showed a more dispersed swimming pattern, moving through all quadrants. Regarding females, the OVX rats showed the best performance on the transfer trial. The HFHF-F and HFHF-OVX rats had a similar behavior and spent the most part of their swimming in the central thirds of the tub in comparison with CF rats, which occurred on the periphery (Figure 3F).

#### 2.2.3. Determination of the Effects of the Hypercaloric Diet on the Short- and Long-Term Memory by Passive Avoidance Task

During the short-term memory test (10 min), no differences were observed among groups (Figure 4A). However, it was noted that the latency of retention of the long-term memory at 24 h decreased in HFHF-M, HFHF-ORCH, and HFHF-F groups compared to their respective control groups (main effect of diet exposure time: F (1, 72) = 49.08, *p* < 0.0001). However, no difference was observed between HFHF-OVX and OVX rats, nor between HFHF-OVS and HFHF-ORCH rats (Figure 4B).

### 2.3. Determination of the Correlation of Oxidative Stress Parameters with the Different Types of Memory

#### 2.3.1. Determination of the Effects of the Hypercaloric Diet on the Oxidative Stress Parameters

When the concentration of MDA was analyzed in the eight experimental groups, increased MDA plasma levels in HFHF-M, ORCH, and HFHF-ORCH groups were reported when compared to the CM group (main effect of diet exposure time ([F (1, 72) = 18.51, *p* < 0.0001], main effect of hormones [F (1, 72) = 12.54, *p* < 0.001]). In females, the HFHF-ORCH group presented a statistically significant increase in MDA concentration compared to CF and HFHF-F groups (main effect of diet exposure time ([F (1, 72) = 18.51, *p* < 0.0001], main effect of hormones [F (1, 72) = 12.54, *p* < 0.001]). Furthermore, MDA plasma levels were higher in HFHF-M rats when compared to HFHF-F animals (main effect of sex [F (1, 72) = 10.22, *p* < 0.01]), but not between the HFHF-ORCH and HFHF-OVX groups (diet × sex × hormones interaction [F (1, 40) = 10.90, *p* < 0.01]) (Figure 5A).

On the other hand, the HFHF-M group showed higher NO plasma levels compared to the CM and ORCH groups (main effect of diet exposure: F (1, 72) = 9.756, *p* < 0.01), diet exposure × hormones interaction: F (1, 72) = 8.223, *p* < 0.01). However, in female animals, no statistical difference was observed between groups (diet × sex × hormones interactions: F (1, 39) = 2.418, *p* > 0.05). Nevertheless, a statistically significant increase in NO concentration was noted in HFHF-M when compared to the HFHF-F group (diet × sex interaction: F (1, 72) = 11.37, *p* < 0.01) (Figure 5B).

#### 2.3.2. Determination of the Correlation of Oxidative Stress Parameters with the Different Types of Memory

Assessment of the Pearson correlation between the MDA or NO concentrations and the NOR results at 24 h showed a negative correlation in HFHF-F and HFHF-OVX groups. However, no other correlations were observed. After the analysis of Pearson correlation between MDA or NO concentrations and the MWM results, a negative correlation between MDA and MWM results was observed in HFHF-M and HFHF-ORCH groups. No other correlation was noted in male groups. Nevertheless, in female rats, a negative correlation between NO and MWM results was observed in the HFHF-OVX group. No correlations were observed for NO nor MDA with the PAT results at 24 h (Table 1).

## 3. Discussion

The consumption of hypercaloric diets is an important factor for the development of MetS both in humans and in animal models that mimic the characteristics of this disease [18,19,20,21]. In accordance with previous studies, it was observed that the intake of a hypercaloric diet increases the body weight, blood pressure, triacylglycerides, and total cholesterol in male rats [2,3,4]. However, most of the studies performed with this type of diet have not evaluated the sex dimorphism of MetS. In addition, these reports show that, as it occurs in humans, male rats are more affected [8,12,13,22,23]. Other studies suggest that male susceptibility to develop MetS may be related to physiological effects of testosterone (T), which is supported by epidemiologic works that have observed that MetS prevalence increases in postmenopausal women in association with an increase in T bioavailability [24].

It should be noted that the hormones affect, in a sex-dependent manner, the parameters of MetS. As in previous studies, in the HFHF-ORCH group, it was noted that the parameters were not modified, while, in ovariectomized female rats, the weight change and concentration of total cholesterol increased even without HFHF diet and, in HFHF-OVX rats, in all metabolic parameters [22,23]. In accordance with the previously reported findings by Do Carmo et al. with regards to the ovariectomy, and with Lee et al. for the orchiectomy, in animals fed with standard diet in the present study, those procedures did not affect the body weight [23,25]. However, an increase in body weight, total cholesterol, and triacylglycerides was found in the HFHF-OVX rats in accordance with Tamaya-Mori et al., and an increase in the systolic blood pressure in accordance with Reckelhoff [22,26]. This may be due to the loss of sex hormones that could induce a redistribution of fat from the subcutaneous to the visceral depot, as well as an impairment of blood pressure due to an increase in vascular resistance or extracellular fluid volume induced by a dysregulation of the renin–angiotensin system [27,28,29,30].

We observed hypertriglyceridemia in HFHF-ORCH and HFHF-OVX compared with the hypercaloric-diet-fed animals. In addition, HFHF-ORCH rats developed hypercholesterolemia when compared with ORCH animals. However, no difference was observed in cholesterol levels between OVX and HFHF-OVX groups. This suggests that sex hormones could play a more relevant role than diet in cholesterol metabolism of female animals. These results are opposite to those observed by Lee et al., as they report that the hypercholesterolemic diet can increase the concentrations of total cholesterol and triacylglycerides in gonadectomized animals but without any statistical difference with respect to the intact animals fed with the same diet [23].

On the other hand, it is known that the behavioral alterations and the cognitive deficit constitute one of the main secondary problems associated with MetS in both humans and animal models [29]. Previous studies have reported effects on memory and learning produced by hypercaloric diets [12,13]. It should be mentioned that the studies conducted by Underwood and Thompson are the only ones performed in both sexes in which the effects on memory and learning were studied through the space object recognition test and the spontaneous alternation test. In these studies, the deterioration of memory and spatial learning (decrease in spontaneous alternation and object recognition index) was observed on intact male and female rats fed with hypercaloric diets [12,13]. In the present study, the rats were first submitted to the object recognition test. Sangüesa et al. reported that only food with fructose and not food with glucose for a long period of time decreases the short- and long-term memory in the NOR task [31]. When evaluating the recognition test of new objects to 2 h, it was noted that most of the groups spent more time with the new object, which is consistent with Arfa-Fatollahkhani et al. [32]. Interestingly, the HFHF-ORCH group presented similar research times to the CF group, and the HFHF-OVX group to the CM group, which would indicate in both cases that the differences in memory could be due to the deficiency of hormones, as previously reported in Karlsson et al. [33]. On the other hand, during assessment of episodic memory at 24 h, the HFHF-M and ORCH groups showed a lower discrimination rate compared to CM and HFHF-ORCH groups, respectively. Previously, Reichelt et al. observed equivalent results and reported similar times on the recognition of the familiar object and the novel object in the NOR test conducted in rats subjected to intermittent hypercaloric diet [34]. Interestingly, the HFHF-ORCH group had an index of object recognition similar to that presented to 2 h, while the group HFHF-OVX did not discriminate between the new object and the familiar one. These results support the influence of gonadal hormones and hypercaloric diet on cognitive functions [35] and are consistent with reports that indicate that the sex hormones: androgens (A), T, and estradiol (E2) can play an important role in the task of object recognition [36,37,38,39]. 

Assessment of spatial learning and memory confirmed other previous reports in which males showed a better spatial learning compared with females, and that a hypercaloric diet decreases latency time of escape in non-gonadectomized groups. We observed similar spatial learning between the HFHF-ORCH and the HFHF-M groups, suggesting that not only hormones have an influence on the components of the spatial memory [40,41]. As previously reported, we observed that the loss of P4 and E2 induced an increase in the escape latency in the OVX group [42]; however, a preventive effect was observed when a hypercaloric diet was administrated. This may be due in part to the synthesis of sex hormones derived from LDL-c and of the excess of acetyl-CoA present in the MetS diet [43,44]. However, further studies are required to determine the molecular mechanisms of these biological/dietary impacts on brain function. Analysis of swimming trajectories during the data transfer phase showed three types of swimming among animal groups: focal, search for scanning, and in circles [45]. During the transfer test, a sex-dependent effect was observed as there was a statistically significant decrease in escape latency in the HFHF-M group compared to the CM group, unlike the increase observed in the HFHF-F, OVX, and HFHF-OVX groups compared to the CF group.

Subsequently, short- and long-term associative memory was evaluated by the passive avoidance test. No differences were observed in short-term memory (10 min) among the groups. Interestingly, all the hypercaloric-fed groups showed a decreased long-term memory (24 h) compared with their respective control groups These data correspond to a previous report by Ganji et.al., in which hypercaloric-fed animals showed a lower retention latency time compared with control fed animals [46]. In addition, in the HFHF-OVX group, no statistical differences were found compared with the OVX group fed with a standard diet as well as what has been observed in the transfer trial in the WMW. This suggests that the hypercaloric diet differentially affects spatial learning and memory in a specific manner. Nevertheless, short- and long-term memory differences were sex-dependent; this variability can influence cognitive performance. Finally, the combination of diet and sex effects, could result in a varied spectrum of cognitive performances.

On the other hand, we further explored the effects of MetS-induced OS on rat behavior by analyzing the correlation between oxidative stress (OS) and memory, and learning, because the visceral adiposity is a key part of the pathogenesis of MetS that results in increased OS, which could be related to behavioral effects in animal models [3,11]. Jiang et al., 2011 reported that hypercaloric diets could increase the activity of the nicotinamide adenine dinucleotide phosphate oxidase (NADPH), which catalyzes the production of superoxide radical by transferring an NADPH electron to oxygen [47]. In this study, we observed increased levels of LPO in hypercaloric-fed animals, except for the HFHF-F group. These results could be related to the antioxidant effects of E2 and pro-oxidant effects of T in HFHF-F animals [48] and supported by the observation of high MDA levels in the HFHF-OVX group and low levels in the HFHF-ORCH group. In accordance with Gancheva, no correlation between the increase in OS and short-term and long-term memory nor spatial learning was observed in HFHF-M or HFHF-ORCH groups studied in the present work [11]. However, in HFHF-M and HFHF-ORCH groups, sex difference in the correlation was found between the increase in the dependent OS and deficits in spatial memory transfer during the test. 

In the case of the determination of nitric oxide (NO), only the HFHF-M rats had increased serum levels. It is known that reactive oxygen species (ROS) can easily react with NO to produce peroxynitrite (ONOO^-^), which inactivates endothelial NO synthase, decreasing the production of NO, resulting in impaired peripheral vasodilation and endothelial dysfunction [49]. Interestingly, a negative correlation between NO levels with long-term recognition memory (NOR 24 h) and spatial memory was observed in the HFHF-OVX group. In addition, a negative correlation between NO levels with long-term recognition memory (NOR 24) was observed in the HFHF-F group. 

The results obtained show the importance of the study on sexual differences of multifactorial pathologies and, in this case, on MetS criteria, as well as in its effects on learning and memory. In addition, the present study shows for the first time, to our knowledge, a sex- and task-type correlation between the systemic OS and different types of memory. However, to understand why these sex- and memory-type-specific differences requires further molecular analysis of neurotrophic factors, such as brain-derived neurotrophic factor (BDNF) or some hormonal receptors that increase the plasticity of the hippocampus and are reduced by different hypercaloric diets in male rats [50,51]. 

### Strengths and Limitations of the Study

Although there are different murine models of MetS, the decision about which model to use for a particular experiment is often multifactorial [18,19]. The advantage of our model over the genetic models is that not the entire population is genetically affected and can develop MetS [18]. Regarding the use of sugars or fat to develop the MetS model, the use of only one of these two elements does not allow the development of all the MetS parameters [18,19]. Another important point is the type of sugar used in the diet. In the case of our model, we decided to use fructose because some studies have reported that the chronic consumption of diets rich in saturated fats and processed sugars (high-fat-and-fructose diet, HFFD), mostly high fructose is strongly associated with a variety of related metabolic diseases, including obesity, systemic insulin resistance, MetS, and type 2 diabetes mellitus [52].

Unfortunately, our country (Mexico) ranks first worldwide in the consumption of soft drinks, which are mostly sweetened with high fructose syrup (60% + 40% sucrose). This is one of the reasons why we decided to use this HFFD. Furthermore, it is essential to mention that the HFFD used in this work substantially reflects the Western hypercaloric diets consisting of high saturated fats and cholesterol, as well as beverages containing high fructose syrup. Because of this, we consider that ours is one of the most complete models of metabolic alterations due to hypercaloric diets [53]. Moreover, this scheme of HFFD has been used previously to produce a MetS model in rats successfully [14,54,55,56].

Although this study is focused on the effect of MetS on memory behavior, one of the limitations of this work is the lack of knowledge of the mechanism through which MetS can affect, at the molecular level, brain structures involved in the regulation of memory and learning processes, such as the hippocampus and the prefrontal cortex. In addition, in a recent work of our group, we reported that a similar diet increases OS and inflammation in the hippocampus [57]. However, further research is required to elucidate these mechanisms in our model.

## 4. Materials and Methods

### 4.1. Animals

In this study, 80 young adult (250–300 g) Sprague–Dawley rats (40 males and 40 females) were obtained from the animal facility of the Hospital de Especialidades, Centro Médico Nacional Siglo XXI. Five animals were kept per cage in acrylic boxes with steady conditions of temperature and humidity, light/dark cycles of 12/12 h (9:00 am–9:00 pm), and water and food (Purina LabDiet^®^ 5008 Richmond, IN, USA) ad libitum. Animals were cared for and handled in compliance with the National Standard NOM-062-ZOO-1999 for the production, care, and use of laboratory animals, for requirements of the National Institutes of Health Guide for the Care and Use of Laboratory Animals (NIH Publication No. 85-23, revised 1985) and the Research Ethics Committee of the Hospital Regional de Alta Especialidad de Ixtapaluca (approval number NR-8-2014). The rats were randomly assigned to one of the following treatment groups: control males (CM), control females (CF), control males with orchidectomy (ORCH), control ovariectomized females (OVX), high-fat and high-fructose [14] fed males (HFHF-M), HFHF-fed females (HFHF-F), HFHF-fed orchiectomized males (HFHF-ORCH), and HFHF-fed ovariectomized females (HFHF-OVX).

### 4.2. Methods

(a)Feeding

The CM, CF, ORCH, and OVX groups were fed with standard diet (3.31 Kcal/g) (*n* = 10 per group). On the other hand, HFHF-M, HFHF-ORCH, HFHF-F, and HFHF-OVX (*n* = 10 per group) were fed with a fat- and fructose-rich diet [14]. The high-fat and high-fructose diet was produced based on standard food (60%), fructose (30%), and pork fat (10%) and it was provided for a period of 12 weeks [14]. To obtain 1 kg of food rich in fat and fructose, 600 g of standard food (LabDiet^®^, Mexico City, Mexico) was pulverized and mixed with 300 g of fructose and 100 g of lard until homogenization (4.161 Kcal/g). Finally, the mixture was cut into pellets such as those of the standard diet.

(b)Gonadectomy

Rats of the ORCH, OVX, HFHF-ORCH, and HFHF-OVX groups were anesthetized with xylazine (10 mg/kg) and ketamine (90 mg/kg). Subsequently, the spinal column of the females and the scrotum in the case of males, were shaved to allow skin exposure. The skin was disinfected with Isodine and animals were placed in right lateral decubitus position.

i.Ovariectomy (OVX). Two lateral dorsal incisions of 1 cm were made to access the peritoneal cavity and locate each ovary. Ligatures with absorbable silk were placed in the oviducts, right below the ovaries, to prevent bleeding. Subsequently, the ovaries were gently removed using sterile scissors, verifying that no bleeding occurred. Following the ovariectomy, oviducts were introduced again in the abdominal cavity. The muscle layer was sutured with absorbable silk and the incisions made in the skin with polypropylene sutures.ii. Orchiectomy (ORCH). Incisions of 1 cm were made in the ventral side of the scrotum to expose the testicles and epididymis. Upon exposition, ligatures with absorbable silk were placed around the blood vessels, and testicle and epididymis were gently removed with sterile scissors, verifying that no bleeding occurred. The remaining content was reintroduced in the testicular sac. The incision on the scrotum was sutured with polypropylene thread.

(c)Body weight, caloric intake, and blood pressure monitoring

The body weight of each animal was recorded at the beginning of the study and weekly for 12 weeks using an electronic weighing scale. Daily caloric consumption was determined weekly. Changes in the systolic blood pressure and heart rate were recorded at the end of the feeding period through a noninvasive method, using an occlusion tail-cuff in the rat’s tail (UgoBasile, Biological Research Apparatus, 21025, Varese, Italy).

(d)Evaluation of memory and learning

All types of memory assessed in the study were evaluated on each animal:

i.Episodic memory. Novel Object Recognition (NOR)

The test was performed in accordance with the previous report by Ennaceur et al. [58], in which the ability of rodents to recognize a new object versus a familiar one is evaluated, reflecting thus the use of learning and recognition memory. The trial was divided into 3 phases: (1) habituation phase; (2) training phase, in which two identical objects were placed for familiarization purposes; and (3) novel object recognition phase, in which a previous familiar object and a completely new one was placed. Each phase was videotaped during 5 min to register exploration time spent for each object. The objects were cleaned with ethanol 85% and were relocated in each test to eliminate any trace left by the previous rodent. The short-term memory was evaluated after 2 h and the long-term memory after 24 h from the beginning of the training. The object discrimination index (DI) was calculated using the formula DI = TN/(TN + TF), in which TN represents the exploration time for the new object and TF the exploration time for the familiar object (TF). A recognition index greater than 0.5 indicates that the episodic memory is not affected [58,59].

ii.Learning and spatial memory. Morris Water Maze (MWM)

This test was performed according to the protocol described by Richard Morris in 1984 to assess the spatial memory in rats. Each animal was submitted to two daily rehearsals, with 20 min of difference between them, for 6 days. The starting point of every trial was different and chosen randomly for each animal. On the seventh day, the spatial memory was evaluated (transfer test) by removing the platform to record the number of times that the rat passed through the place previously occupied by the platform. Each test lasted 60 s and was videotaped. Swimming trajectories and escape latency were obtained using the recorded images to evaluate spatial learning [60].

iii.Short- and long-term memory. Passive avoidance test (PAT).

The test was performed with the protocol previously reported by Pinto-Almazán et al., 2014 [61]. In this test, the animal was conditioned with an aversive stimulus and, subsequently, it was assessed whether or not it remembered that experience. The first retention test was composed of two sessions, one of 10 min (short-term memory) and one of 24 h (long-term memory). In the first session, the animal was trained by placing it in the safety compartment and allowing it to enter the punishment compartment. Once in the punishment compartment, the animal received a 3 mA electric shock for 3 s, and then it was allowed to leave the punishment compartment and was isolated for 10 min in its home cage. Subsequently, the short-term memory was evaluated by putting the animal into the safety compartment and observing if the animal entered the punishment compartment before the 600 s (total test duration). If the animal entered before the 600 s, the test was concluded and the latency was recorded. The long-term memory was assessed using the same procedure 24 h later and the latency time was recorded.

(e)Sacrifice and blood collection

After the behavioral tests were conducted, animals were decapitated. Blood samples were collected in a test tube and centrifuged at 1372× *g* for 15 min at 4 °C. The serum was collected and stored at −80 °C until the completion of the analysis.

(f)Triacylglycerides and cholesterol analysis

Serum samples were thawed and both triacylglycerides and cholesterol were determined through triacylglycerides and cholesterol spectrophotometric assays GPO (Teco Diagnostics, Anaheim, CA, USA) by enzyme-linked immunosorbent assay following the manufacturer’s instructions. The concentration was calculated using the following formula: Cs = (As/Astd) * CStd, where Cs: concentration of the sample expressed in mg/dL, As: absorbance of the sample, AStd: absorbance of the standard, and CStd: concentration of the standard 200 mg/dL [62].

(g)Determination of lipid peroxidation

Lipid peroxidation was assessed by measuring MDA using the thiobarbituric acid (TBA)-reactive substances test [63] and was calculated in µmol/L. A volume of 200 µL of 25% trichloroacetic acid was added to an aliquot of plasma. The samples were incubated at 4 °C for 15 min, followed by centrifugation at 4 °C, 5000× *g* for 3 min. Supernatants (100 µL) were neutralized with a 4 M NaOH solution (JT Baker, Xalostoc, Edo. De Mexico, Mexico). Then, 1 mL of 0.7% TBA (Acros Organics, Geel, Belgium) solution was added to the neutralized supernatant (final dilution 1:1). This mixture was incubated at 90 °C for 60 min. The color reaction was measured spectrophotometrically (532 nm) in the organic phase (1-butanol, Sigma-Aldrich, St. Louis, MO, USA). Tetramethoxypropane was used as a standard. The results were expressed as TBARs per µmol/L. 

(h)Determination of Nitric Oxide (NO)

The nitric oxide was measured by determining the total quantity of nitrite (NO_2_^-^), which is the stable product of NO metabolism in serum. The Griess reagent was used (an aqueous solution of 1% sulfanilamide (Sigma-Aldrich, St. Louis, MO, USA) with 0.1% naphthylethylenediamine (Sigma-Aldrich, St. Louis, MO, USA) in 2.5% H_3_PO_4_ (2.5%, JT Baker, Xalostoc, Mexico), which forms a stable chromophore with NO_2_^-^ and absorbs light at 546 nm [64]). The calibration curve was constructed using different concentrations of sodium nitrite dissolved in 0.9% NaCl. The NO levels were expressed in µmol/L.

(i)Statistical analysis

The data are presented as means ± standard deviation (SD). Statistical analysis of food consumption, ELISA technique, and behavioral results between treatment groups were calculated using a three-way ANOVA with the post hoc Tukey test (GraphPad Prism version 8 for Windows, GraphPad Software, La Jolla, CA, USA). The statistical significance for all data was established at *p* < 0.05, *n* = 10 for comparison between groups. 

## 5. Conclusions

These results suggest that the changes observed in the different types of memories and learning evaluated in the present study, and those that have been previously reported to be affected by MetS, are sex- and hormone-dependent in the animals fed with HFHF. Moreover, the results indicate that the effects in the types of memory that are impaired might by associated to the increase in oxidative stress markers. 

## Figures and Tables

**Figure 1 metabolites-12-00341-f001:**
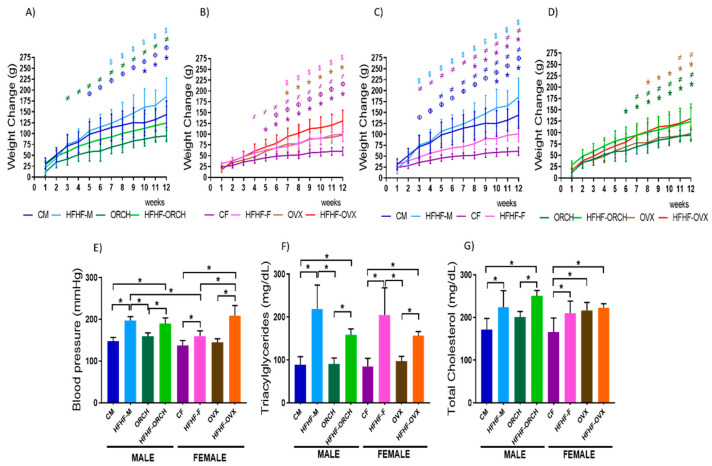
Effects of the HFHF diet on (**A**–**D**) weight, (**E**) blood pressure, (**F**) triacylglycerides, and (**G**) cholesterol. Weight change produced by HFHF in (**A**) male, (**B**) female, (**C**) non-gonadectomized, and (**D**) gonadectomized groups. Changes induced by HFHF diet in (**E**) blood pressure, (**F**) triacylglycerides, and (**G**) cholesterol in all experimental groups. The HFHF groups present greater body weight gain, increased blood pressure, increased concentration of triacylglycerols and cholesterol than animals fed with standard diet. (*n* = 10 per group.) CM: control males, HFHF-M: metabolic syndrome males, ORCH: males with orchidectomy, HFHF-ORCH: metabolic syndrome males with orchidectomy, CF: control females, HFHF-F: metabolic syndrome females, OVX: females with ovariectomy, and HFHF-OVX: metabolic syndrome females with ovariectomy. The graphs represent the mean ± SD. (*n* = 10.) Evidence of 3-way ANOVA with post hoc Tukey test for weight. * *p* < 0.05 (CM vs. HFHF-M, ORCH vs. HFHF-ORCH, CF vs. HFHF-F and OVX vs. HFHF-OVX) *n* = 10, Φ *p* < 0.05 (CM vs. ORCH, CF vs. OVX, CM vs. CF and ORCH vs. OVX), $ *p* < 0.05 (HFHF-M vs. HFHF-ORCH, HFHF-F vs. HFHF-OVX, HFHF-M vs. HFHF-F and HFHF-ORCH vs. HFHF-OVX) and ≠ *p* < 0.05 (CM vs. HFHF-ORCH, ORCH vs. HFHF-M, CF vs. HFHF-OVX, OVX vs. HFHF-F, CM vs. HFHF-F, CF vs. HFHF-M, ORCH vs. HFHF-OVX and OVX vs. HFHF-ORCH). The color of the symbol indicates the group compared. Body weight: main effect of diet exposure [F (1, 72) = 36.03 *p* < 0.0001]; main effect of sex [F (1, 72) = 43.48 *p* < 0.0001]; main effect of hormones [F (1, 72) = 3.095, *p* = 0.0830]; and interaction of sex and hormones [F (1, 72) = 55.54, *p* < 0.0001]. Blood pressure: main effect of diet exposure [F (1, 72) = 134.6, *p* < 0.0001]; main effect of sex [F (1, 72) = 9.277, *p* < 0.05]; main effect of hormones [F (1, 72) = 18.04, *p* < 0.0001]; and interaction of sex and hormones [F (1, 72) = 55.54, *p* < 0.0001]; and diet exposure, sex, × hormone interaction [F (1, 72) = 17.96, *p* < 0.0001]. Hypertriglyceridemia: main effect of diet exposure: [F (1, 72) = 97.22, *p* < 0.0001]; main effect of hormones [F (1, 72) = 10.36, *p* < 0.05]; sex × hormone interaction [F (1, 72) = 4.499, *p* < 0.05]; and diet exposure × hormone interaction [F (1, 72) = 18.57, *p* < 0.001]. Hypercholesterolemia: main effect of diet exposure [F (1, 72) = 29.59 *p* < 0.0001]; main effect of hormones [F (1, 72) = 18.15, *p* < 0.001].

**Figure 2 metabolites-12-00341-f002:**
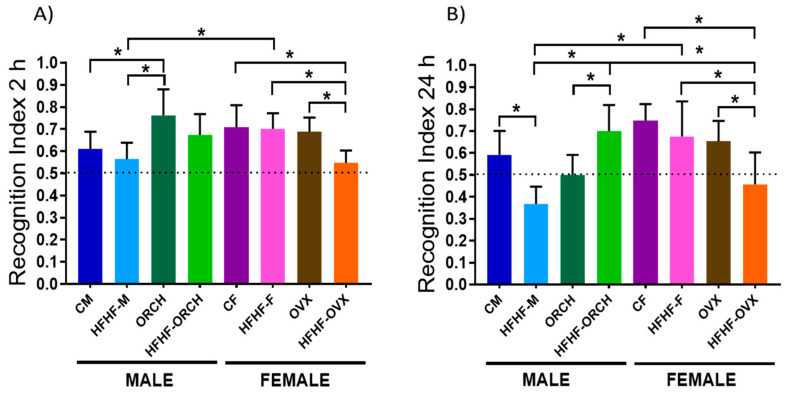
Effect of the HFHD diet in the recognition test of new objects. (**A**) The episodic memory in short time to 2 h is not affected in any of the experimental groups. (**B**) To 24 h, the HFHF-M group has a greater tendency to the familiar object. The dotted line represents that, below 0.5, the rat spent more time with the familiar object, while above 0.5 spent more time with the new object (*n* = 10 per group). CM: control males, HFHF-M: metabolic syndrome males, ORCH: males with orchidectomy, HFHF-ORCH: metabolic syndrome males with orchidectomy, CF: control females, HFHF-F: metabolic syndrome females, OVX: females with ovariectomy, and HFHF-OVX: metabolic syndrome females with ovariectomy. The graphs represent the mean ± SD. Evidence of 3-way ANOVA with post hoc Tukey test. * *p* < 0.05 between the groups. Recognition index 2 h: main effect of diet exposure [F (1, 72) = 10.88, *p* < 0.01]; sex × hormones interaction [F (1, 72) = 25.65, *p* < 0.0001]; and diet exposure × hormone interaction [F (1, 72) = 4.280, *p* < 0.05]. Recognition index 24 h: main effect of diet exposure [F (1, 72) = 6.224, *p* < 0.05]; main effect of sex [F (1, 72) = 10.29, *p* < 0.01]; diet exposure × sex interaction [F (1, 72) = 4.454, *p* < 0.05]; sex × hormones interaction of [F (1, 72) = 22.37, *p* < 0.0001]; and diet exposure, sex, × hormone interaction [F (1, 72) = 22.00, *p* < 0.0001].

**Figure 3 metabolites-12-00341-f003:**
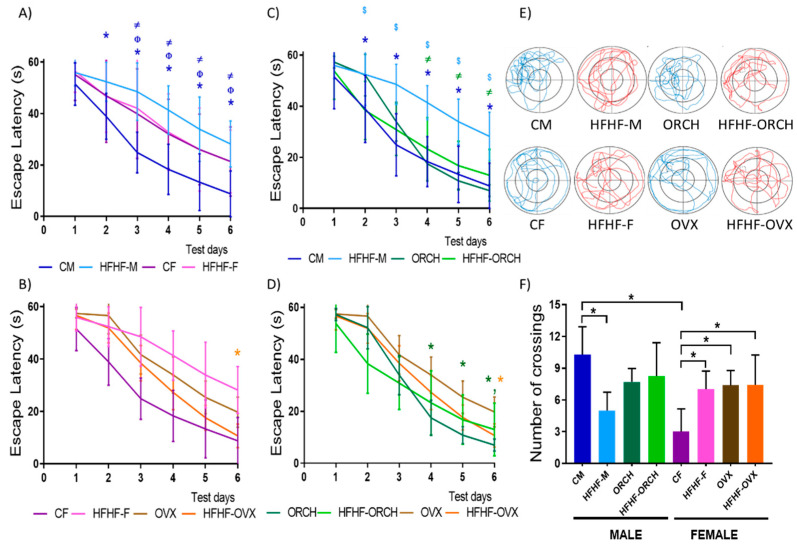
Effect of the HFHF diet in the tests of spatial learning and memory. (**A**) Escape latency in males: non-orchiectomized fed with standard diet, orchiectomized fed with standard diet, non-orchiectomized fed with standard diet, and orchiectomized fed with HFHF diet. The HFHF-M group presented a greater escape latency compared to CM, ORCH, and HFHF-ORCH groups. (**B**) Escape latency in females: non-ovariectomized fed with standard diet, ovariectomized fed with standard diet, non-ovariectomized fed with standard diet, and ovariectomized fed with HFHF diet. The HFHF-OVX group presented a smaller escape latency compared to CF, OX, and HFHF-OVX groups. (**C**) Escape latency in males and females: non-gonadectomized fed with standard diet and non- gonadectomized fed with HFHF diet. The HFHF-M, CF, and HFHF-F groups had greater escape latency compared to CM animals. (**D**) Escape latency in males and females: gonadectomized fed with standard diet and gonadectomized fed with HFHF diet. HFHF-ORCH presented a greater escape latency than ORCH group and HFHF-OVX had a lower escape latency compared to OVX group. (**E**) Representative trajectories in the transfer trial of aquatic labyrinth of Morris. The groups HFHF-M, HFHF-F, and OVX toured greater distance than the other study groups. (**F**) Transfer trial of the aquatic labyrinth of Morris. (*n* = 10 per group.) CM: control males, HFHF-M: metabolic syndrome males, ORCH: males with orchiectomy, HFHF-ORCH: metabolic syndrome males with orchiectomy, CF: control females, HFHF-F: metabolic syndrome females, OVX: females with ovariectomy, and HFHF-OVX: metabolic syndrome females with ovariectomy. The data represent the mean ± SD. Evidence of 3-way ANOVA with post hoc Tukey test for escape latency. (**A**–**D**) * *p* < 0.05 (CM vs. HFHF-M, ORCH vs. HFHF-ORCH, CF vs. HFHF-F and OVX vs. HFHF-OVX) *n* = 10, Φ *p*<0.05 (CM vs. ORCH, CF vs. OVX, CM vs. CF and ORCH vs. OVX), $ *p* < 0.05 (HFHF-M vs. HFHF-ORCH, HFHF-F vs. HFHF-OVX, HFHF-M vs. HFHF-F and HFHF-ORCH vs. HFHF-OVX) and ≠ *p* < 0.05 (CM vs. HFHF-ORCH, ORCH vs. HFHF-M, CF vs. HFHF-OVX, OVX vs. HFHF-F, CM vs. HFHF-F, CF vs. HFHF-M, ORCH vs. HFHF-OVX and OVX vs. HFHF-ORCH). The color of the symbol indicates the group compared. The color of the symbol indicates the group compared. Spatial Learning 6th day: main effect of diet exposure [F (1, 72) = 10.02, *p* < 0.01]; main effect of sex [F (1, 72) = 4.056, *p* < 0.05]; main effect of hormones [F (1, 72) = 30.02, *p* < 0.0001]; diet exposure × sex interaction [F (1, 72) = 35.87, *p* < 0.0001]; diet exposure × hormones interaction [F (1, 72) = 11.93, *p* < 0.001]. F) Evidence of 3-way ANOVA with post hoc Tukey test for transfer trial of the aquatic labyrinth of Morris. * *p* < 0.05. Spatial Memory 7th day: main effect of sex [F (1, 72) = 5.796, *p* < 0.05]; main effect of hormones [F (1, 72) = 6.534, *p* < 0.05]; and diet exposure, sex, × hormone interaction [F (1, 72) = 14.36, *p* < 0.001].

**Figure 4 metabolites-12-00341-f004:**
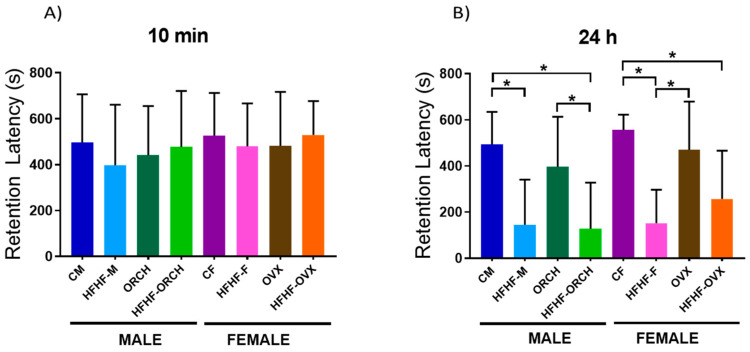
Effect of HFHF diet on short- and long-term memory. (**A**) Retention latency in short-term memory (10 min) of all studied groups. (**B**) Retention latency in the long-term memory (24 h) was observed to decline in the latency of the HFHF-M, HFHF-ORCH, and HFHF-F groups vs. their peers. CM: control males, HFHF-M: metabolic syndrome males, ORCH: males with orchidectomy, HFHF-ORCH: metabolic syndrome males with orchidectomy, CF: control females, HFHF-F: metabolic syndrome females, OVX: females with ovariectomy, and HFHF-OVX: metabolic syndrome females with ovariectomy. The data represent the mean ± SD. Evidence of 3-way ANOVA with post hoc Tukey test. * *p* < 0.05. Retention Latency 24 h: main effect of diet exposure [F (1, 72) = 49.08, *p* < 0.0001].

**Figure 5 metabolites-12-00341-f005:**
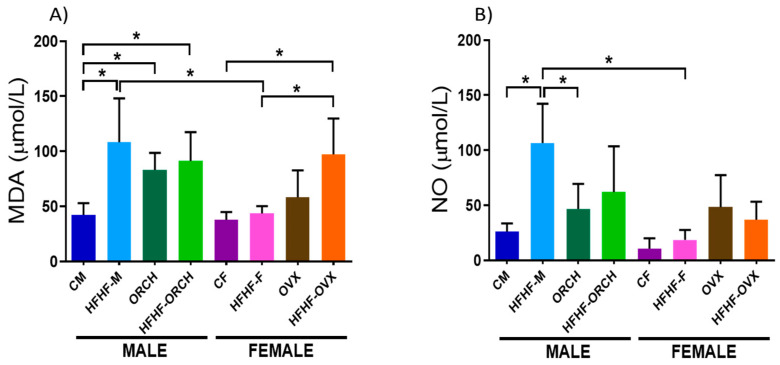
Effect of HFHF diet on oxidative stress markers. (**A**) Effects of HFHF diet on the concentration of MDA. (**B**) Effects of HFHF diet on the concentration of NO. CM: control males, HFHF-M: metabolic syndrome males, ORCH: males with orchidectomy, HFHF-ORCH: metabolic syndrome males with orchidectomy, CF: control females, HFHF-F: metabolic syndrome females, OVX: females with ovariectomy, and HFHF-OVX: metabolic syndrome females with ovariectomy. The graphs represent the mean ± SD. Evidence of 3-way ANOVA with post hoc Tukey test. * *p* < 0.05. MDA: main effect of diet exposure [F (1, 72) = 18.51, *p* < 0.0001]; main effect of sex [F (1, 72) = 10.22, *p* < 0.01]; main effect of hormones [F (1, 72) = 12.54, *p* < 0.001]; and diet exposure, sex, × hormone interaction [F (1, 72) = 10.90, *p* < 0.01]. NO: main effect of diet exposure [F (1, 72) = 9.756, *p* < 0.01]; main effect of sex [F (1, 72) = 18.56, *p* < 0.0001]; and diet exposure × sex interaction [F (1, 72) = 11.37, *p* < 0.01] and sex × hormone interaction [F (1, 72) = 7.403, *p* < 0.01] and diet exposure × hormone interaction [F (1, 72) = 8.223, *p* < 0.01].

**Table 1 metabolites-12-00341-t001:** Results of the Pearson correlation between the concentration of MDA and NO with the results of 24 h NOR, MWM, and PAT.

Cognitive Evaluation	Oxidative Stress Biomarkers	HFHF-M		HFHF-ORCH		HFHF-F		HFHF-OVX	
		Pearson r	*p*	Pearson r	*p*	Pearson r	*p*	Pearson r	*p*
NOR 24 h	MDA	−0.2869	0.5327	−0.5211	0.2304	0.5118	0.2403	−0.1877	0.6869
	NO	−0.1752	0.7071	0.0526	0.9108	−0.7724	0.0418 *	−0.9054	0.0130 *
MWM	MDA	−0.7711	0.0424 *	−0.7560	0.0493 *	−0.1971	0.6719	0.4432	0.3192
	NO	−0.3128	0.4946	0.5641	0.1871	−0.1581	0.7349	−0.8632	0.0268 *
PAT 24 h	MDA	0.0721	0.8873	0.1112	0.8405	0.3604	0.4262	−0.4643	0.3024
	NO	0.3571	0.4444	0.4077	0.3714	0.1081	0.8222	0.2029	0.7000

Pearson correlation between the different types of memory and the concentrations of oxidative stress biomarkers was calculated. High negative correlation was found between MWM and MDA in HFHF-M and HFHF-ORCH groups. High negative correlation between NOR 24 h and NO in HFHF-F animals and very high negative correlation in HFHF-OVX groups. High correlation between MWM and NO was found. “*”, *p* < 0.05.

## Data Availability

The data presented in this study are available on request from the corresponding authors. The data are not publicly available due to other ongoing investigations and manuscripts.

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
