# Peer review of "High Fructose and High Fat Diet Impair Different Types of Memory through Oxidative Stress in a Sex- and Hormone-Dependent Manner"

_metabolites, 2022, doi:10.3390/metabo12040341_

Round 1

Reviewer 1 Report

This manuscript is interesting and potentially useful to understand the influence sexual differences on multifactorial pathologies related to MetS, particularly regarding to its effects on learning and memory. Although mechanisms are not determined, authors need to discuss the main advantages and limitations of this study regarding to other animal models.  

Author Response

Answers to Reviewer 1 concerns:

This manuscript is interesting and potentially useful to understand the influence sexual differences on multifactorial pathologies related to MetS, particularly regarding to its effects on learning and memory.

We are thankful for the time and effort you have invested in the revision of our manuscript. Your suggestions have enriched our work. In the manuscript, the additions are highlighted in yellow. Please find our answers to your valuable recommendations; we hope that we have addressed your concerns. 

  1. Although mechanisms are not determined, authors need to discuss the main advantages and limitations of this study regarding to other animal models.

Answer:

Thank you for your kind suggestion. We have included in the section 3. Discussion, subsection: 3.1. Strengths and limitations of the study.

3.1 Strengths and limitations of the study

Although there are different murine models of MetS, the decision about which model to use for a particular experiment is often multifactorial [18,19]. The advantage of our model over the genetic models is that not the entire population is genetically affected and can develop MetS [18]. Regarding the use of sugars or fat to develop the MetS model, the use of only one of these two elements does not allow the development of all the MetS parameters [18,19]. Another important point is the type of sugar used in the diet. In the case of our model, we decided to use fructose because some studies have reported that the chronic consumption of diets rich in saturated fats and processed sugars (high-fat-and-fructose diet, HFFD), mostly high fructose, is strongly associated with a variety of related metabolic diseases including obesity, systemic insulin resistance, MetS, and type-2 diabetes mellitus [52].

Unfortunately, our country (Mexico) ranks first worldwide in the consumption of soft drinks, which are mostly sweetened with high fructose syrup (60% + 40% sucrose). This is one of the reasons why we decided to use this HFFD. Furthermore, it is essential to mention that the HFFD used in this work substantially reflects the Western hypercaloric diets consisting of high saturated fats and cholesterol, as well as beverages containing high fructose syrup. Because of this, we consider that ours is one of the most complete models of metabolic alterations due to hypercaloric diets [53]. Also, this scheme of HFFD has been used previously to produce a MetS model in rats successfully [14,54–56].

Although, this study is focused on the effect of MetS on memory behavior, one of the limitations of this work is the lack of knowledge of the mechanism through which MetS can affect, at the molecular level, brain structures involved in the regulation of memory and learning processes, such as the hippocampus and the prefrontal cortex. In addition, in a recent work of our group, we reported that a similar diet increases OS and inflammation in the hippocampus [57]. However, further research is required to elucidate these mechanisms in our model.

Reviewer 2 Report

The manuscript metabolites-1665368 entitled “High fructose and high fat diet impair different types of memory through oxidative stress in a sex- and hormone-dependent manner” by Edwin Chávez-Gutiérrez and coworkers studies the changes in body weight, blood pressure, blood lipids, oxidative stress markers and alterations in different types of memory in Sprague-Dawley rats fed with an enriched fructose and fat (HFHF) diet were evaluated.

After 12 weeks of feeding the male and female rats with HFHF increases the body weight, blood pressure and generate dyslipidemia in comparison to the animals fed with chow diet. Regarding the memory, it was noted that gonadectomy reverted the effects of HFHF in the 24h Novel Object Recognition Task and in the spatial learning/memory analyzed through Morris Water Maze being more affected males than females. These results suggest that the changes observed in the memory and learning of MetS animals are sex- and hormone dependent and correlated to increase in oxidative stress.

The design of the experimental work is fine.

Methodology is appropriated.

Results are clearly presented, figures are meaningful.

The discussion is consistent with results.

Major observations:

Line 591: results are reported as mean ± SEM. Actually the proper measure of dispersion should be standard deviation. SEM is used for inferential statistics, otherwise in the case of this work, it is like to reduce the SD by 3.3 times. This is actually not statistically correct.

Author Response

The manuscript metabolites-1665368 entitled “High fructose and high fat diet impair different types of memory through oxidative stress in a sex- and hormone-dependent manner” by Edwin Chávez-Gutiérrez and coworkers studies the changes in body weight, blood pressure, blood lipids, oxidative stress markers and alterations in different types of memory in Sprague-Dawley rats fed with an enriched fructose and fat (HFHF) diet were evaluated.

After 12 weeks of feeding the male and female rats with HFHF increases the body weight, blood pressure and generate dyslipidemia in comparison to the animals fed with chow diet. Regarding the memory, it was noted that gonadectomy reverted the effects of HFHF in the 24h Novel Object Recognition Task and in the spatial learning/memory analyzed through Morris Water Maze being more affected males than females. These results suggest that the changes observed in the memory and learning of MetS animals are sex- and hormone dependent and correlated to increase in oxidative stress.

The design of the experimental work is fine.

Methodology is appropriated.

Results are clearly presented, figures are meaningful.

The discussion is consistent with results.

We appreciate the time and effort you have invested in the revision of our manuscript. Indeed, all your suggestions have improved the quality of our manuscript. In the main text, the additions are highlighted in yellow. We hope that we have correctly addressed all your concerns.

 (x) English language and style are fine/minor spell check required

Answer:

We are sorry for this unintentional situation. We have checked the manuscript using commercial software (Grammarly) and have asked an English native speaker for proofreading. 

  1. Major observations: Line 591: results are reported as mean ± SEM. Actually, the proper measure of dispersion should be standard deviation. SEM is used for inferential statistics, otherwise in the case of this work, it is like to reduce the SD by 3.3 times. This is actually not statistically correct.

 Answer:

Thank you for your valuable comments. We have changed all the Figures graphs included in the section 3. Results in each subsection.

Reviewer 3 Report

For convenience of readers, it could be useful to include a graph or scheme summarizing the aims of the study.

In figure captions, authors should include the ANOVA P values.

Author Response

Answers to Reviewer 3 concerns:

We appreciate the time and effort you have invested in the revision of our manuscript. Indeed, all your suggestions have improved the quality of our manuscript. In the main text, the additions are highlighted in yellow. We hope that we have correctly addressed all your concerns.

  1.  (x) English language and style are fine/minor spell check required

Answer:

We are sorry for this unintentional situation. We have checked the manuscript using commercial software (Grammarly) and have asked an English native speaker for proofreading. 

  1. For convenience of readers, it could be useful to include a graph or scheme summarizing the aims of the study.

 Answer:

Thank you for your suggestion. We believe that the Graphical abstract would help to catch the readers' attention of the study and improve the readability of the text. 

  1. In figure captions, authors should include the ANOVA P values.

Answer:

Thank you for your valuable comments. We have included a more detailed description in the Figures Captions to present the ANOVA P values in the graphs included in the section 3. Results in each subsection.

Figure 1. Effects of the HFHF diet on weight, triacylglycerols, cholesterol and blood pressure. The HFHF groups present greater body weight gain, increased blood pressure, in the concentration of triacylglycerols and cholesterol than animals fed with standard diet. (n=10 per group). CM: control males, HFHF -M: metabolic syndrome males, ORCH: males with orchidectomy, HFHF-ORCH: metabolic syndrome males with orchidectomy, CF: control females, HFHF-F: metabolic syndrome females, OVX: females with ovariectomy and HFHF-OVX: metabolic syndrome females with ovariectomy. The graphs represent the mean ± SD. (n=10). Evidence of 3-way ANOVA with post hoc Tukey test for weight. Body weight: Main effect of diet exposure [F (1,72) = 36.03 p <0.0001]; main effect of sex [F (1,72) = 43.48 p <0.0001]; main effect of hormones [F (1,72) = 3.095, p = 0.0830]; and interaction of sex and hormones [F (1,72) = 55.54, p <0.0001]. Blood pressure: Main effect of diet exposure [F (1, 72) = 134.6, p <0.0001]; main effect of sex [F (1, 72) = 9.277, p <0.05]; main effect of hormones [F (1,72) = 18.04, p <0.0001]; and interaction of sex and hormones [F (1,72) = 55.54, p <0.0001] and Diet exposure, sex, x hormone interaction [F (1, 72) = 17.96, p <0.0001]. Hypertriglyceridemia: Main effect of diet exposure: [F (1, 72) = 97.22, p <0.0001]; main effect of hormones [F (1,72) = 10.36, p <0.05]; Sex x hormone interaction [F (1,72) = 4.499, p <0.05]and Diet exposure x hormone interaction [F (1,72) = 18.57, p <0.001]. Hypercholesterolemia: Main effect of diet exposure [F (1,72) = 29.59 p <0.0001]; main effect of hormones [F (1,72) = 18.15, p <0.001].

Figure 2. Effect of the HFHD diet in the recognition test of new objects. A) The episodic memory in short time to 2 h is not affected in any of the experimental groups. B) To the 24 h the group HFHF-M has a greater tendency by the familiar object. The dotted line represents below the 0.5 that the rat was more time with the familiar object, above the 0.5 spent more time with the new object (n=10 per group). CM: control males, HFHF -M: metabolic syndrome males, ORCH: males with orchidectomy, HFHF-ORCH: metabolic syndrome males with orchidectomy, CF: control females, HFHF-F: metabolic syndrome females, OVX: females with ovariectomy and HFHF-OVX: metabolic syndrome females with ovariectomy. The graphs represent the mean ± SD. Evidence of 3-way ANOVA with post hoc Tukey test. *p<0.05.  Recognition index 2h: Main effect of diet exposure [F (1, 72) = 10.88, p <0.01]; and sex x hormones interaction [F (1,72) = 25.65, p <0.0001] and Diet exposure x hormone interaction [F (1, 72) = 4.280, p <0.05]. Recognition index 24h: Main effect of diet exposure [F (1, 72) = 6.224, p <0.05]; main effect of sex [F (1, 72) = 10.29, p <0.01]; diet exposure x sex interaction [F (1,72) = 4.454, p <0.05]; sex x hormones interaction of [F (1,72) = 22.37, p <0.0001] and Diet exposure, sex, x hormone interaction [F (1, 72) = 22.00, p <0.0001].

Figure 3. Effect of the HFHF diet in the tests of spatial learning and memory. A) Escape latency in males: non-orchiectomized fed with standard diet, orchiectomized fed with standard diet, non-orchiectomized fed with standard diet and orchiectomized fed with HFHF diet. The HFHF-M group presented a greater escape latency compared to CM, ORCH and HFHF-ORCH groups. B) Escape latency in females: non-ovariectomized fed with standard diet, ovariectomized fed with standard diet, non-ovariectomized fed with standard diet and ovariectomized fed with HFHF diet. The HFHF-OVX group presented a smaller escape latency compared to CF, OX and HFHF-OVX groups. C) Escape latency in males and females: non-gonadectomized fed with standard diet and non- gonadectomized fed with HFHF diet. The HFHF-M, CF and HFHF-F groups had greater escape latency compared to CM animals. D) Escape latency in males and females: gonadectomized fed with standard diet and gonadectomized fed with HFHF diet. HFHF-ORCH presented a greater escape latency than ORCH group and HFHF-OVX had smaller escape latency compared to OVX group. E) Representative trajectories in the transfer trial of aquatic labyrinth of Morris. The groups HFHF-M, HFHF-F and OVX toured greater distance than the other study groups. F) Transfer trial of the aquatic labyrinth of Morris. (N=10 per group) CM: control males, HFHF -M: metabolic syndrome males, ORCH: Males with orchiectomy, HFHF-ORCH: metabolic syndrome males with orchiectomy, CF: control females, HFHF-F: metabolic syndrome females, OVX: females with ovariectomy and HFHF-OVX: metabolic syndrome females with ovariectomy. The data represent the mean ± SD. Evidence of 3-way ANOVA with post hoc Tukey test for escape latency. A-D) *p <0.05, Φp<0.05, $p<0.05 and ≠p<0.05. The color of the symbol indicates the group compared. Spatial Learning 6th day: Main effect of diet exposure [F (1, 72) = 10.02, p <0.01]; Main effect of sex [F (1, 72) = 4.056, p <0.05]; main effect of hormones [F (1,72) = 30.02, p <0.0001]; diet exposure x sex interaction [F (1, 72) = 35.87, p <0.0001]; diet exposure x hormones interaction [F (1,72) = 11.93, p <0.001]. F) Evidence of 3-way ANOVA with post hoc Tukey test for transfer trial of the aquatic labyrinth of Morris. *p <0.05. Spatial Memory 7th day: Main effect of sex [F (1, 72) = 5.796, p <0.05]; main effect of hormones [F (1,72) = 6.534, p <0.05]; and diet exposure, sex, x hormone interaction [F (1, 72) = 14.36, p <0.001].

Figure 4. Effect of HFHF diet in short and long-term memory. A) Retention latency in short-term memory (10 min) of all studied groups. B) Retention latency in the long-term memory (24 h) was observed decline in the latency of the HFHF -M, HFHF-ORCH and HFHF-F groups vs against their peers. CM: control males, HFHF -M: metabolic syndrome males, ORCH: males with orchidectomy, HFHF-ORCH: metabolic syndrome males with orchidectomy, CF: control females, HFHF-F: metabolic syndrome females, OVX: females with ovariectomy and HFHF-OVX: metabolic syndrome females with ovariectomy. The data represent the mean ± SD. Evidence of 3-way ANOVA with post-hoc Tukey test. *p<0.05. Retention Latency 24h: Main effect of diet exposure [F (1, 72) = 49.08, p <0.0001].    

Figure 5. Effect of HFHF diet on oxidative stress markers. A) Effects of HFHF diet on the concentration of MDA. B) Effects of HFHF diet on the concentration of NO. CM: control males, HFHF -M: metabolic syndrome males, ORCH: males with orchidectomy, HFHF-ORCH: metabolic syndrome males with orchidectomy, CF: control females, HFHF-F: metabolic syndrome females, OVX: females with ovariectomy and HFHF-OVX: metabolic syndrome females with ovariectomy. The graphs represent the mean ± SD. Evidence of 3-way ANOVA with post hoc Tukey test. *p<0.05. MDA: Main effect of diet exposure [F (1, 72) = 18.51, p <0.0001]; main effect of sex [F (1, 72) = 10.22, p <0.01]; main effect of hormones [F (1,72) = 12.54, p <0.001]; and diet exposure, sex, x hormone interaction [F (1, 72) = 10.90, p <0.01]. NO: Main effect of diet exposure [F (1, 72) = 9.756, p <0.01]; main effect of sex [F (1, 72) = 18.56, p <0.0001]; and diet exposure x sex interaction [F (1, 72) = 11.37, p <0.01] and sex x hormone interaction [F (1, 72) = 7.403, p <0.01] and diet exposure x hormone interaction [F (1, 72) = 8.223, p <0.01].
